# Platelet-Derived Extracellular Vesicles as Lipid Carriers in Severe Allergic Inflammation

**DOI:** 10.3390/ijms241612714

**Published:** 2023-08-12

**Authors:** Alba Couto-Rodriguez, Alma Villaseñor, Carmela Pablo-Torres, David Obeso, María Fernanda Rey-Stolle, Héctor Peinado, José Luis Bueno, Mar Reaño-Martos, Alfredo Iglesias Cadarso, Cristina Gomez-Casado, Coral Barbas, Domingo Barber, María M. Escribese, Elena Izquierdo

**Affiliations:** 1Departamento de Ciencias Médicas Básicas, Instituto de Medicina Molecular Aplicada (IMMA) Nemesio Díez, Facultad de Medicina, Universidad San Pablo-CEU, CEU Universities, Urbanización Montepríncipe, 28660 Boadilla del Monte, Spain; alba.coutorodriguez@beca.ceu.es (A.C.-R.);; 2Centro de Metabolómica y Bioanálisis (CEMBIO), Facultad de Farmacia, Universidad San Pablo-CEU, CEU Universities, Urbanización Montepríncipe, 28660 Boadilla del Monte, Spain; 3Spanish National Cancer Research Center (CNIO), Molecular Oncology Programme, Microenvironment and Metastasis Laboratory, 28029 Madrid, Spain; 4Department of Hematology, Hospital Universitario Puerta de Hierro Majadahonda, 28222 Madrid, Spain; 5Department of Allergy and Immunology, Hospital Universitario Puerta de Hierro Majadahonda, 28222 Madrid, Spain

**Keywords:** extracellular vesicles, platelets, lipids, inflammation, allergy, metabolomics

## Abstract

The resolution of inflammation is a complex process that is critical for removing inflammatory cells and restoring tissue function. The dysregulation of these mechanisms leads to chronic inflammatory disorders. Platelets, essential cells for preserving homeostasis, are thought to play a role in inflammation as they are a source of immunomodulatory factors. Our aim was to identify key metabolites carried by platelet-derived extracellular vesicles (PL-EVs) in a model of allergic inflammation. PL-EVs were isolated by serial ultracentrifugation using platelet-rich plasma samples obtained from platelet apheresis from severely (*n* = 6) and mildly (*n* = 6) allergic patients and non-allergic individuals used as controls (*n* = 8). PL-EVs were analysed by a multiplatform approach using liquid and gas chromatography coupled to mass spectrometry (LC-MS and GC-MS, respectively). PL-EVs obtained from severely and mildly allergic patients and control individuals presented comparable particle concentrations and sizes with similar protein concentrations. Strikingly, PL-EVs differed in their lipid and metabolic content according to the severity of inflammation. L-carnitine, ceramide (Cer (d18:0/24:0)), and several triglycerides, all of which seem to be involved in apoptosis and regulatory T functions, were higher in PL-EVs from patients with mild allergic inflammation than in those with severe inflammation. In contrast, PL-EVs obtained from patients with severe allergic inflammation showed an alteration in the arachidonic acid pathway. This study demonstrates that PL-EVs carry specific lipids and metabolites according to the degree of inflammation in allergic patients and propose novel perspectives for characterising the progression of allergic inflammation.

## 1. Introduction

The inflammatory response is a complex defence mechanism triggered in response to injury, infection, or tissue damage, which includes a resolving phase with counter-regulatory factors, the apoptosis of inflammatory cells, and the initiation of healing [1,2]. However, inflammation may become chronic, contributing to a variety of long-term diseases such as cancer, diabetes, or allergy [3,4]. Inflammatory diseases cause the death of three out of five people and their prevalence is expected to increase continuously over the next 30 years [3]. Nonetheless, the cellular and molecular mechanisms underlying chronic inflammatory disorders remain unknown.

Recent studies have highlighted platelet alterations in inflammatory conditions such as rheumatoid arthritis [5], multiple sclerosis [6], Crohn’s disease [7], sepsis [8], and, more recently, allergy [9,10]. In fact, we have previously found that platelets from severely allergic patients present a specific lipid content that could contribute to the development of chronic inflammation [10]. In these inflammatory disorders, platelets become activated and release their content together with extracellular vesicles (EVs) [11,12], which could alter platelet–immune cell communication. However, the exact contribution of EVs to the progression and perpetuation of inflammation is still unknown.

EVs are small double-membrane vesicles that cannot replicate and are secreted by all cells. They contain a rich molecular cargo, including proteins, lipid mediators, and microRNAs that can be transferred to target cells, thus playing an important role in intercellular communication [13,14,15]. Platelet-derived EVs (PL-EVs) represent the highest percentage of EVs in blood [16,17] and have been shown to exhibit immunomodulatory actions on various cell types [12] such as endothelial cells [18] and leukocytes [19], including T cells [20] and monocytes [21]. In addition, PL-EVs can influence the local microenvironment through transcellular lipid metabolism, like the delivery of arachidonic acid to other cells, which induces COX-2 expression and prostacyclin production [18]. Likewise, PL-EVs contain lysophosphocholines (LPCs), a main lipid inducing platelet activation, aggregation, adhesion, spreading, and migration, as well as platelet–monocyte aggregate formation [22]. Although lipids are key molecules in the inflammatory response and major components of EVs, the metabolic content of PL-EVs under inflammatory conditions remains unexplored.

In this study, using a model of respiratory allergic inflammation stratified according to severity [10], the metabolic load of PL-EVs was investigated, aiming to provide insights into novel mediators and mechanisms involved in the progression of inflammation into a severely allergic condition.

## 2. Results

### 2.1. Patient Cohort and Platelet Sample Characterisation

The clinical history of recruited patients was carefully analysed, and all subjects (*n* = 20) were classified into three groups according to the criteria outlined above: non-allergic individuals used as controls (*n* = 8) and mildly allergic (*n* = 6) and severely allergic (*n* = 6) patients. There were no differences among the three groups in terms of gender, age, and smoking habits (*p* > 0.05). Moreover, the age of onset of rhinoconjunctivitis and asthma reactions was unable to differentiate between the mild and severe allergic groups (*p* > 0.05) or their sensitisation profile. However, there were statistically significant differences between the mild and severe groups in forced vital capacity (FVC) and forced expiratory volume in one second (FEV1) (*p* < 0.05), with the severe group showing the lowest values (less than 80%) in 83.3% of severely allergic patients. More information on the groups can be found in Appendix A.

Whole blood hemograms showed no differences in platelet and white blood cell counts between the experimental groups. As we have previously shown [10], platelet-rich plasma (PRP) hemograms confirmed the absence of other blood cell types, which ensured the purity of the PL-EV samples. Likewise, platelet counts and mean platelet volume did not differ between experimental groups (*p* > 0.05, Appendix A). Therefore, any further differences observed would be due to the severity of their allergic profile rather than their haematological or demographic profile.

### 2.2. Characterisation of PL-EVs in Patients with Different Allergic Inflammatory Degrees

The purification of EVs is challenging and downstream assay results vary depending on the technique used. In the absence of a standardised protocol, two techniques, serial ultracentrifugation (UC) and size exclusion chromatography followed by ultrafiltration (SEC + UF), were tested for the purification of PL-EVs from controls (Appendix A). PL-EVs were characterised according to the MISEV2018 guideline indications [13]. First, a transmission electron microscope (TEM) showed the purity and morphology of the PL-EVs (Figure 1A). Particle number and size were analysed by nanoparticle tracking analysis (NTA), observing that the samples extracted by SEC + UF were more homogeneous, although there were no significant differences between the methods (*p* > 0.05) (Figure 1B–D). The protein:particle ratio was also measured, obtaining similar results between both purification methods (Figure 1E,F and Appendix A). Finally, the expression of different EV markers was analysed (Figure 1G and Appendix A) and showed that PL-EVs expressed EV-specific markers (ALIX, CD63, and CD9), as well as the CD61 marker, mainly expressed on platelets. The presence of proteins not associated with EVs, such as albumin and apoproteins, was also observed. In summary, the isolation of PL-EVs by both methods was successful and the purified EVs were shown to belong to platelets. The UC method was selected beacause it allows direct resuspension of the PL-EV pellet with the buffer required for the subsequent analyses.

Next, we characterised the size and concentration of PL-EVs from mild and severe allergic patients and controls (Figure 1H–L and Appendix A). Statistical analyses showed no significant differences between any of the phenotypes in terms of particle number, size, protein concentration, or protein:particle ratio (*p* > 0.05). Therefore, these results ensure that any differences found in subsequent studies will only be due to their cargo.

### 2.3. Comparative Metabolic Analysis of PL-EVs from Patients with Different Allergic Inflammatory Grades

For the metabolomic analyses, PL-EVs obtained from control subjects and mild and severe allergic patients were measured by liquid and gas chromatography coupled to mass spectrometry (LC-MS and GC-MS, respectively) platforms. To ensure the quality of the metabolomic data, non-supervised models using Principal Component Analysis (PCA) models were carried out (Appendix A). A tight cluster of the quality control samples (QCs) was observed in the graphs proving the good quality of the data, meaning that the differences that are later shown by the groups are indeed of a biological nature.

Then, non-supervised analyses again using PCA models were conducted for the PL-EVs data between the groups (Figure 2). The PCA models revealed that the extreme groups, PL-EVs obtained from severely allergic patients and control subjects (Figure 2, upper row), were more separated compared to mildly allergic patients and controls (Figure 2, middle row). However, our data showed only a slight–moderate trend in the PL-EV’s metabolic cargo between mildly and severely allergic patients (Figure 2, lower row). Here, the PCA model from LC-MS in negative mode (ESI−) showed a slightly better clustering of both groups that was not observed in LC-MS in positive mode (ESI+) and GC-MS. This points out that PL-EV metabolites that charge negatively might classify better than other metabolites from the ESI+ or from other techniques such as GC-MS.

Subsequently, to look for differences between the groups, discriminant analysis models using partial least square discriminant analysis (PLS-DA) were conducted by pair of groups (Figure 3). We observed that the best quality parameters in terms of the sample classification (R^2^) and prediction (Q^2^) scores of the PLS-DA models were obtained by comparing severely allergic patients with controls (LC-MS in negative ionisation mode (ESI−) model: R^2^ = 0.74; Q^2^ = 0.62; GC-MS model: R^2^ = 0.60; Q^2^ = 0.51; Figure 3, upper row), Likewise, the PLS-DA model from mildly allergic patients compared to controls also exhibited good scores (LC-MS ESI− model: R^2^ = 0.78; Q^2^ = 0.50; Figure 3, middle row). Finally, when comparing the allergic groups (severe vs. mild) the PLS-DA model showed a moderate classification score but a moderate–low prediction quality (LC-MS in positive ionisation mode (ESI+) model: R^2^ = 0.68; Q^2^ = 0.24; Figure 3, lower row). Thus, our data suggest a link between the metabolic load of PL-EVs and the inflammatory status of allergic patients.

### 2.4. Defining the Lipid and Metabolic Changes in the PL-EV Cargo in Allergic Patients Compared to Control Subjects

Once we stated that the main metabolic differences were between the mild and severe allergic groups against controls, we investigated which signals were involved by performing a univariate analysis. Figure 4A shows that the comparison of the extreme groups (control subjects vs. severely allergic patients) had the highest number of significant differences (*p* < 0.05; *p*-FDR < 0.1) with 225 chemical signals, whereas the comparison of control vs. mild and mild vs. severe presented 81 and 17 significant signals, respectively. Significant metabolites for each pairwise comparison were used for constructing heatmaps (Figure 4B–D). The hierarchical clustering analyses showed that when comparing controls with mildly and severely allergic patients, only one allergic patient was misclassified within the control group, meaning 80% correct classification (Figure 4B,C). Of note, both allergy groups presented a decreased abundance of most significant metabolites in PL-EVs when compared to the control group (Figure 4B,C). To a lesser extent, hierarchical clustering classified allergic patients according to their inflammatory degree (Figure 4D). Therefore, the metabolic content of PL-EVs from mildly and severely allergic patients is different from controls, showing that allergic patients have primarily lower levels of lipids in comparison to control subjects.

### 2.5. Identifying Metabolites Carried by PL-EVs in Patients with Allergic Inflammation

Given the distinct differences observed among the PL-EVs of control subjects and mild and severe allergic patients, we ought to annotate for the LC-MS data (ESI− and ESI+) the significant chemical signals that characterise the PL-EVs of each group. After annotation by fragmentation experiments, a total of 30 metabolites were identified among the three comparisons, including LC-MS (ESI− and ESI+) and GC-MS (Appendix A).

The identified metabolites were mainly lipids classified as lysophospholipids (LPLs), phospholipids (PLs), glycerolipids (GLs), and sphingolipids (SLs). Additionally, metabolites such as hydroxyurea and L-carnitine were identified (Figure 5A,B and Appendix A). Notably, most of the LPLs and PLs, as well as the sphingomyelin SM (34:1), hydroxyurea, and phosphoric acid, were decreased in PL-EVs from mild and severe allergic patients compared to the control group (Figure 5A and Appendix A).

On the other hand, L-carnitine, a key metabolite in fatty acid oxidation (FAO), was highly abundant in mildly allergic patients’ PL-EVs compared to both the control group (*p* < 0.01, *p*-FDR = 0.112; Figure 5A,B and Appendix A) and the severe allergic group (*p*= 0.056, *p*-FDR = 0.672; Figure 5 and Appendix A and Appendix A). Regarding severe allergic patients, our data showed that their PL-EVs were more enriched in sphingosine (*p* < 0.05, *p*-FDR < 0.01) and monoglyceride MG (18:0) (*p* < 0.05, *p*-FDR = 0.117) compared to the control group and had higher levels of phosphatidylcholine PC (34:1) (*p* < 0.05, *p*-FDR < 0.05) than the mild patients (Figure 5A). On the contrary, PL-EVs from severe allergic patients showed reduced levels of the ceramide Cer (d18:1/20:4) (*p* < 0.05, *p*-FDR < 0.05) and specific triglycerides (TGs) (*p* < 0.05, *p*-FDR < 0.05) compared to the mildly allergic group (Figure 5A,B and Appendix A). Therefore, PL-EVs from allergic patients are loaded with specific lipid and metabolic mediators according to the patient’s inflammatory grade.

### 2.6. Biological Pathways Associated with PL-EV Metabolomic Profiles in Allergic Inflammation

Considering the metabolic alterations identified in PL-EVs from allergic patients with different degrees of inflammation, an enrichment pathway analysis was performed with the identified significant metabolites to explore the biological routes involved. After an analysis of the three comparisons, significant pathways (*p* < 0.05) in developmental biology, metabolism, and signal transduction were identified (Figure 5C and Appendix A). Specifically, PL-EVs obtained from mildly and severely allergic patients compared to controls, showed alterations in pathways related to small molecule transport, in particular the ABC family of transporters, and pathways involved in the de novo synthesis of SL, which play an important role in cell proliferation and apoptosis (Appendix A). Also, the arachidonic acid metabolism and phospho-lipase A2 (PLA2) pathways were shown to be affected routes in severely allergic patients when compared to controls (Appendix A). Likewise, PL-EVs obtained from patients with severely allergic inflammation showed alterations in immune system pathways compared to the control and mild groups (Figure 5C), indicating their possible role as metabolic mediators. Specifically, different phagocytosis-related pathways, such as Fc gamma receptor-dependent phagocytosis (FCGR) and the role of PL in phagocytosis, were found to be significant (Appendix A).

Overall, our results suggest that PL-EVs are a source of metabolic mediators that play an important immunoregulatory role associated with allergic inflammation.

## 3. Discussion

Allergic inflammation is a complex physiological process involving various cells and molecules of the immune system [23], where platelets play an important role [24,25,26], carrying diverse lipid mediators that vary according to the grade of inflammation [26,27]. In addition, platelets release EVs that, unlike platelets, can cross endothelial barriers and extend their reach and potential impact to other fluids and tissues [11]. However, the role of PL-EVs in the inflammatory response remains unclear.

In this exploratory study, we demonstrate that PL-EVs exhibit a differential metabolic cargo according to the degree of inflammation in a well-established model of inflammatory allergy stratified by severity. Moreover, the observed metabolic changes suggest that PL-EVs might participate in important biological pathways involved in the regulation of the inflammatory response.

Currently, the processing of EVs is a great challenge due to the lack of standardised purification methods, which results in different outcomes depending on the technique used [13]. Our data showed that purification of PL-EVs from PRP samples by the two most commonly used methods (UC and SEC + UF) produced similar results in terms of particle number and size and protein expression and quantity. We also characterised PL-EVs obtained from homeostatic (control subjects) and inflammatory conditions (mild and severe allergic patients) and found no differences in the protein:particle ratio, particle number, and particle size between the different phenotypes. These results seem contrary to those previously described in the literature for other inflammatory diseases such as rheumatoid arthritis [28], multiple sclerosis [29], lupus erythematosus [30], and asthma [31], where increased particle numbers were described in patients with these diseases. This is probably because these studies included all sorts of platelet-derived particles such as larger EVs (e.g., microparticles and apoptotic bodies) and small EVs (e.g., exosomes), as samples were not subjected to high-speed UC steps [19,32]. In this work, we specifically studied small PL-EVs, obtained by the serial UC (100,000× *g*) of PRP samples. Importantly, these results mean that it is not the particle size or number of PL-EVs, but variances in lipid and metabolite abundance that are responsible for the differences found in subsequent PL-EV load analyses.

Next, aiming to determine the specific metabolic profile of PL-EVs in allergic patients, we performed a multiplatform analysis by LC-MS and GC-MS on PL-EV samples from each experimental group. These analyses resulted in the annotation of several significant lipids and metabolites that determined metabolic differences for each phenotype: control, mildly allergic, and severely allergic patients. To our knowledge, there are no previous data comparing the metabolic load of PL-EVs among inflammatory diseases or between patients with different degrees of inflammation. On the other hand, the metabolic content of platelets obtained from patients with inflammation has been better studied [33]. Recent work on allergic inflammation revealed that platelet lipid content is different according to severity, where severely allergic patients’ platelets presented high levels of Cer, phosphoinositols, phosphocholines, and SMs [10]. Now, our results show that the lipid and metabolite content of PL-EVs allowed for the separation of mild and severe allergic patients from control subjects, with a clearer separation between the extreme groups (severe vs. controls). As a whole, we were able to classify patients by their degree of inflammation according to their metabolic PL-EV content.

Additionally, after identifying the metabolic changes in the PL-EV load, the associated metabolic pathways were identified. We found that compared to controls, PL-EVs from mild and severe allergic patients have lower levels of the PLs and LPLs included in Figure 5A, which are sources of fatty acids used to generate energy and act as precursors of proinflammatory mediators. In contrast, both, specific PLs and LPLs have been shown to be more abundant in platelets from allergic patients than from controls [10], suggesting that these metabolites could be retained in platelets, whereas in controls they are released via PL-EVs.

Also, PL-EVs from mild allergic patients were highly loaded with Cer (d18:1/24:0) compared to PL-EVs from severely allergic patients and control subjects. Although we only see one ceramide, it has been shown that ceramides may play an important role in inflammation. Nonetheless, platelets obtained from severe allergic patients display a higher abundance of ceramides [10], which might indicate that in the severe phenotypes, ceramides could be retained in platelets, whereas in mild allergic patients they are released via PL-EVs. Ceramides are considered central components in sphingolipid metabolism [34,35] and their accumulation activates the migration, phagocytosis, and production of inflammatory cytokines [34]. In addition, ceramides inhibit the mitochondrial respiratory chain, generating reactive oxygen species (ROS) and inducing cell apoptosis [36]. The fact that mildly allergic patients’ PL-EVs contain higher levels of Cer (d18:1/24:0) could mean that apoptotic signals are sent through PL-EVs to control the inflammatory response in the immune system. This result seems to be supported by the higher abundance of specific TGs also detected in mildly allergic patients’ PL-EVs, as an increased concentration of TGs can trigger apoptosis [37,38]. We can therefore hypothesise that PL-EVs from patients with mild inflammation might regulate the immune response by removing inflammatory cells, which is not the case for patients with severe inflammation. Additionally, TGs can promote the differentiation of T cells into regulatory T cells (Tregs), as was shown in cancer and transgenic mouse models [39,40]. Interestingly, PL-EVs derived from mildly allergic patients also showed elevated levels of L-carnitine, an essential metabolite in the transport of fatty acids into the mitochondria carried out by FAO. Increased oxidative phosphorylation (OXPHOS) and FAO generate reactive oxygen species (ROS) promoting FoxP3 signalling, which potentiates the suppressive function of Tregs [41]. Hence, the elevated levels of L-carnitine and TG in PL-EVs from mildly allergic patients might suggest that regulatory signals to control inflammation are being transmitted to target inflammatory cells. On the other hand, these metabolic signals are drastically lower in PL-EVs from severely allergic patients compared to mildly allergic patients.

Furthermore, PL-EVs obtained from patients with severely allergic inflammation exhibited the dysregulation of metabolites involved in PLA2 enzyme activity and arachidonic acid metabolism, key metabolic pathways implicated in the modulation of the inflammatory response, whose alterations have been previously reported in several inflammatory disorders, including severely allergic inflammation [9,10,42].

Altogether, our results suggest that PL-EVs are a source of metabolic mediators that could participate in several biological pathways involved in the regulation of apoptosis and Treg functions essential in the progression of allergic diseases, which might affect the resolution of the inflammatory response. Although further research is required to validate the functional relevance of these findings, our results provide new insights into the underlying mechanisms of allergic inflammation.

## 4. Materials and Methods

### 4.1. Patients

Twenty individuals (aged 18–55 years) were recruited between October 2018 and February 2021. The protocol was approved by the “Hospital Universitario Puerta De Hierro” (HUPHM), Madrid, Spain, Research and Ethics Committees, and written informed consent was obtained from all subjects. Eight subjects were healthy and non-allergic (tested by Skin Prick Test; SPT) and were taken as controls. The remaining subjects were allergic patients recruited at the Allergy Service of the HUPHM [10].

Inclusion criteria for allergic patients were a clinical history of allergy to aeroallergens proved by SPT. Allergic patients were stratified by severity into mildly and severely allergic groups according to GINA guidelines [43]. Six severe patients (GINA step 5) met at least one of the following criteria: (1) poor asthma control as assessed by ACT (Asthma Control Test) < 20 or ACQ (Asthma Control Questionnaire) > 1.5; (2) two or more severe exacerbations/two or more glucocorticosteroid courses for more than three days each (within the previous year); (3) one or more hospitalisations for a severe exacerbation (within the previous year). The six remaining patients were included in the mild group (GINA 1–4). Patients younger than 18 years or with concomitant inflammatory diseases, cancer, or haematological diseases were excluded.

### 4.2. Sample Collection and Processing

Briefly, as previously described [10], platelet apheresis was performed at the Apheresis Unit (haematology department) of the HUPHM. The plateletpheresis machine was set to obtain 85 mL of PRP using Adenine Citrate Dextrose-A (ACD-A) as an anticoagulant during the process. PRP samples were collected under sterile conditions. After a resting period of 2 h, a hemogram was performed to evaluate cell counts. The quality requirement for PRP was set at a minimum concentration of 500 × 10^9^ platelets/L. The presence of other cell types was found to be negligible.

### 4.3. Isolation Methods for EVs

As it has been shown that measurements of plasma- and blood-derived EVs depend on several variables (sample source, anticoagulant, storage conditions, etc.), and there are no standardised methods for isolating PL-EVs from plasma-apheresis samples [13], we applied the two most commonly used methods for EV purification: ultracentrifugation and size exclusion chromatography followed by ultrafiltration.

#### 4.3.1. Ultracentrifugation (UC)

Two millilitres of PRP sample was centrifuged at 700× *g* for 7–10 min at room temperature (RT). The supernatant fraction was taken and centrifuged at 12,000× *g* for 20 min at 10 °C. The supernatant was then diluted in phosphate-buffered saline (PBS) and ultracentrifuged using a Type 55.2 TI. SER#17557 rotor (Optima L-90K ultracentrifuge; Beckman Coulter, Fullerton, CA, USA) at 100,000× *g* for 70 min at 10 °C. An amount of 3 mL of PBS was added to the pellet and ultracentrifuged using an SW 55 Ti SER#97U 2526 rotor (Optima L-90K ultracentrifuge; Beckman Coulter, Fullterton, CA, USA) at 100,000× *g* for 70 min at 10 °C. The supernatant fraction was removed. The tube was placed upside down on absorbent paper to remove excess PBS. Finally, the pellet containing the EVs was resuspended in 100 µL of PBS (10010023, Thermo Fisher Scientific, Waltham, MA, USA) for EV Characterisation or 50 µL of cold mass spectrometry (MS)-grade methanol (MeOH) (047192.K2, Thermo Fisher Scientific, Waltham, MA, USA) for metabolomic analysis.

#### 4.3.2. Size Exclusion Chromatography and Ultrafiltration (SEC + UF)

SEC was performed with qEV2/35nm columns and according to the manufacturer’s instructions (Izon Science, Ltd., Christchurch, New Zealand). Briefly, 2 mL of PRP was centrifuged at 500× *g* for 7–10 min at RT and then the supernatant was centrifuged at 12,000× *g* for 20 min at 10 °C. The supernatant was loaded onto a qEV2/35nm column. The EV-containing fractions were collected and concentrated by Amicon^®^ Ultra-2 Centrifugal Filter Devices (UFC200324, Millipore, Germany) following the manufacturer’s instructions.

### 4.4. EV Characterisation

#### 4.4.1. Protein Quantification

Protein concentration was measured using the BCA Protein Assay Kit (23225, Thermo Fisher Scientific, Waltham, MA, USA) following the manufacturer’s protocol. Proteins were stored at −80 °C until use.

#### 4.4.2. Western Blot

Quantified proteins were aliquoted and boiled with Laemmli Sample Buffer (#1610747, Biorad, Hercules, CA, USA) for 5 min at 95 °C. The sample was separated on 12% SDS-PAGE under reducing conditions (Laemmli Buffer with 5% 2-Mercaptoethanol (#1610710, Biorad, Hercules, CA, USA) or without reduction, according to the manufacturer’s antibody instructions. The transfer was carried out on a Nitrocellulose Membrane, 0.2 µm (#1620112, Biorad, Hercules, CA, USA). Membranes were incubated overnight with primary antibodies: Anti-ALB/Albumin (sc-271605, Santa Cruz Biotechnology, Santa Cruz, CA, USA), Anti-Apolipoprotein A1 (sc-376818, Santa Cruz Biotechnology, Santa Cruz, CA, USA), Anti-Apolipoprotein B (ab139401, Abcam, Cambridge, UK), Anti-TSG101 (ab125011, Abcam, Cambridge, UK), Anti-CD9 (ab58989, Abcam, Cambridge, UK), Anti-CD81 (sc-166029, Santa Cruz Biotechnology, Santa Cruz, CA, USA), anti-CD63 Monoclonal (Ts63) (Thermo Fisher Scientific, Waltham, MA, USA), Anti-ALIX (ab88743, Abcam, Cambridge, UK), and Anti-Integrin beta 3 (CD61) (ab179473, Abcam, Cambridge, UK) at 4 °C. The antibodies were chosen in agreement with MISEV2018 guidelines [13]. Clarity ECL Western Blotting Substrate (Biorad, Hercules, CA, USA) was used to visualise bands and was revealed in the ChemiDoc Imaging System (Biorad, Hercules, CA, USA).

#### 4.4.3. Nano Tracking Particle Analysis (NTA)

EV samples were analysed by NTA using a NanoSight NS500 system equipped with a violet laser (405 nm) (Malvern Panalytical Ltd. Worcestershire, UK). For each sample, three 60 s videos were recorded at a camera level of 12–14. The temperature was monitored during the recording. The recorded videos were analysed at a detection threshold of 10 using NTA Software (version NTA 3.4, 2018) (Malvern Panalytical Ltd. Worcestershire, UK) to determine the concentration and size distribution of the measured particles along with the corresponding standard error. For optimal measurements, samples were diluted with PBS until the particle concentration was within the optimal concentration range for particle analysis (1 × 10^8^–1 × 10^9^).

#### 4.4.4. Transmission Electron Microscopy (TEM)

EV isolates were prepared at a concentration of 0.1 μg/μL in PBS with 2% PFA. Five μL of the suspension was deposited on a layer of Parafilm©. A formvar/carbon-coated grid (TEM-FCF200CU50, Sigma Aldrich, Darmstadt, Germany) was placed over the drop and left to adsorb for 20 min. The grid was washed with drops of PBS. Subsequently, the grid was placed on a drop of 50 μL of 1% glutaraldehyde (16216 EMS, Hatfield, PA, USA) in PBS and allowed to fix for 5 min. The grid was washed with drops of distilled water. For contrast and inclusion, the grid was placed on a drop of uranyl oxalate and allowed to contrast for 5 min. Subsequently, the grid was placed on a drop of methyl cellulose–uranyl acetate on ice. The remaining reagent was absorbed with Whatman paper and allowed to dry at RT.

### 4.5. Statistical Analysis

All raw data produced were tested for normality using the Shapiro–Wilk test and expressed as the mean ± SD. For paired and non-paired 2-group comparisons, the Wilcoxon and Mann–Whitney U tests were used, respectively. For comparisons of three groups, the ANOVA or Kruskal–Wallis tests were used according to the normality of the data Differences were considered statistically significant when *p*-values were less than 0.05 (*), 0.01 (**) and 0.001 (***). All the statistical analyses were performed using SPSS (version 27.0.1, 2021) (SPSS Inc.; Chicago, IL, USA) and graphical presentations were created using GraphPad Prism 8 (version 8.4.2, 2020) (GraphPad Software Inc.; San Diego, CA, USA).

### 4.6. Metabolomic Studies

PL-EVs samples were analysed by a multiplatform using LC-MS and GC-MS. For LC-MS, an Agilent HPLC system (1290 infinity II series) coupled with a quadrupole time-of-flight analyser system (Q-ToF MS 6545) (Agilent Technologies, Santa Clara, CA, USA) was used. Additionally, for GC-MS, an Agilent GC system (1890 series) equipped with an autosampler MultiPurpose Sampler (MPS, Gerstel, Germany) coupled with a single quadrupole gas chromatography/mass spectrometry system (GC MSD 5977B series) was employed. Both techniques followed previously described methodologies [44,45,46]. Full descriptions of QC preparation, detailed instrumental description, data treatment, metabolite annotation, and statistical analysis are available in the Appendix A.

#### 4.6.1. Sample Treatment for LC-MS

EVs were purified from 5 mL of PRP sample by UC (as described above). The samples were mixed by vortexing for 30 s and sonicated for 10 min. Then, 25 µL was transferred to another tube and 5 µL of SPLASH lipidomics internal standard mix (SPLASH^®^ LIPIDOMIX^®^ Mass Spec Standard, 330707, Avanti Polar Lipids, Inc. Alabaster, AL, USA) was added to each of the samples, mixed by vortexing for 30 s, and sonicated for 10 min. The sediment was removed by centrifugation at 16,000× *g* for 20 min at 4 °C, and 25 µL from the supernatant was transferred to an LC vial for analysis. A second extraction was performed. Thus, another 25 µL of cold methanol was added to the pellet. The samples were mixed by vortexing, sonicated for 10 min, and centrifuged again under the same conditions. From the supernatant, 15 µL was added to the sample’s corresponding LC vial for analysis. The order of the analysis of all samples was randomised before metabolite extraction.

#### 4.6.2. Sample Treatment for GC-MS

From the remaining sample of PL-EVs, previously extracted in methanol and analysed by LC-MS, 20 μL was transferred to a new GC vial with an insert. Then 25 μL of palmitic acid D-31 at 50 ppm in acetonitrile was added and evaporated to dryness (SpeedVac Concentrator, Thermo Fisher Scientific, Waltham, MA, USA). Next, 10 μL of O-methoxyamine hydrochloride in pyridine (15 mg/mL) was added to each GC vial, and the mixture was vigorously vortex-mixed and ultrasonicated. Methoxymation was carried out in darkness, at room temperature for 16 h. Afterwards, 10 μL of N,O-Bis(trimethylsilyl)trifluoroacetamide (BSTFA) with 1% trimethylchlorosilane (TMCS) as a catalyst was added and the solution was further mixed using the vortex. For the silylation process, samples were heated in an oven for 1 h at 70 °C. Finally, 50 μL of heptane containing 20 ppm of tricosane (IS) was added to each GC vial and vortex mixed before GC analysis.

## 5. Conclusions

PL-EVs show differences in their metabolic content between inflammatory allergic patients and control subjects. In mild allergic patients, the Cer (d18:1/24:0), specific TGs and L-carnitine are released through PL-EVs, while these metabolites are less abundant in severely allergic patients’ PL-EVs. These metabolites are involved in metabolic and inflammatory signalling pathways, including apoptosis and the regulatory response by Tregs. Thus, PL-EVs from mildly allergic patients may regulate inflammation and eliminate inflammatory cells, whereas, in severely allergic patients, we observed dysregulation in inflammation-associated metabolites. In summary, the results of this pilot study suggest that PL-EVs from allergic patients are a source of metabolites involved in immune regulation, which might affect the resolution of the inflammatory response.

## Figures and Tables

**Figure 1 ijms-24-12714-f001:**
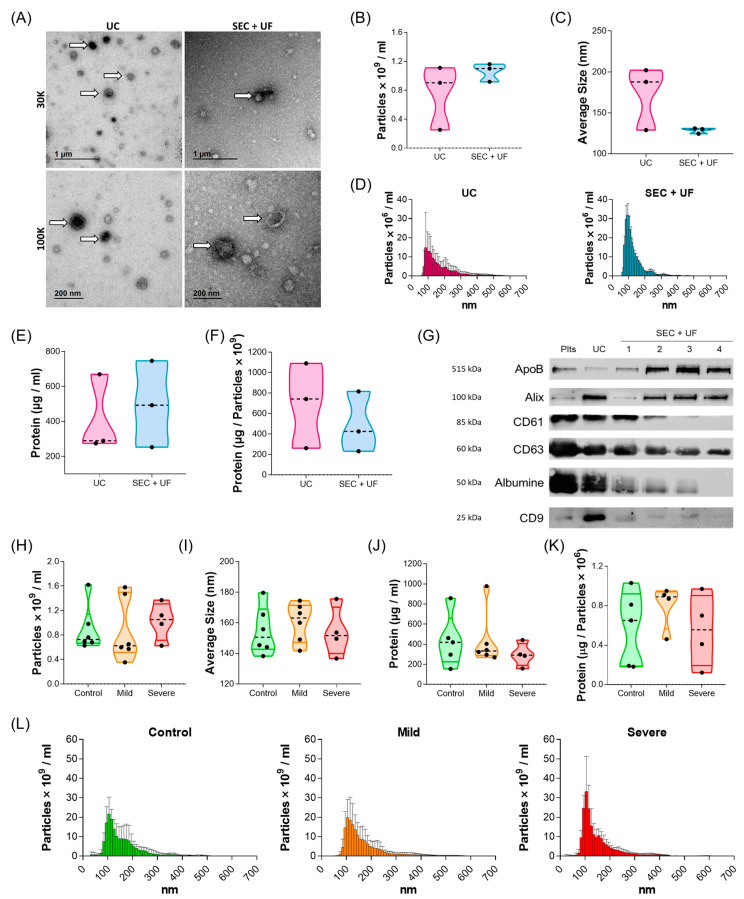
Characterisation of PL-EVs. (**A**–**G**) PL-EVs isolated from control subjects (*n* = 3): (**A**) Representative TEM images from UC and SEC + UF extraction. Thirty thousand (30K) and 100,000 (100 K) magnifications are shown, white arrows indicate some of the PL-EVs presented in the samples. (**B**) Quantification of the number of particles/mL. (**C**) Size (diameter, mean) of particles and (**D**) size distribution of particles determined by NTA. (**E**) Quantification of total protein cargo and (**F**) ratio of protein per particle. (**G**) Western blot results from a control subject for platelet marker (CD61), EV markers (ALIX, CD63, CD9), and non-EV components (ApoB, Albumin). The numbers corresponding to the SEC + UF represent the first four fractions collected. (**H**–**L**) PL-EVs isolated by UC from controls and mild and severe patients: (**H**) number of particles/mL and (**I**) size (diameter, mean) of particles measured by NTA (*n* = 5–6); (**J**) particle protein concentration (*n* = 4–5); (**K**) ratio of protein:particle (*n* = 4–5); and (**L**) size distribution of particles determined by NTA (*n* = 5–6). Plts, platelets; UC, ultracentrifugation method; SEC + UF, size exclusion chromatography + ultrafiltration method; ApoB, Apoprotein B; ALIX, ALG-2 interacting protein X.

**Figure 2 ijms-24-12714-f002:**
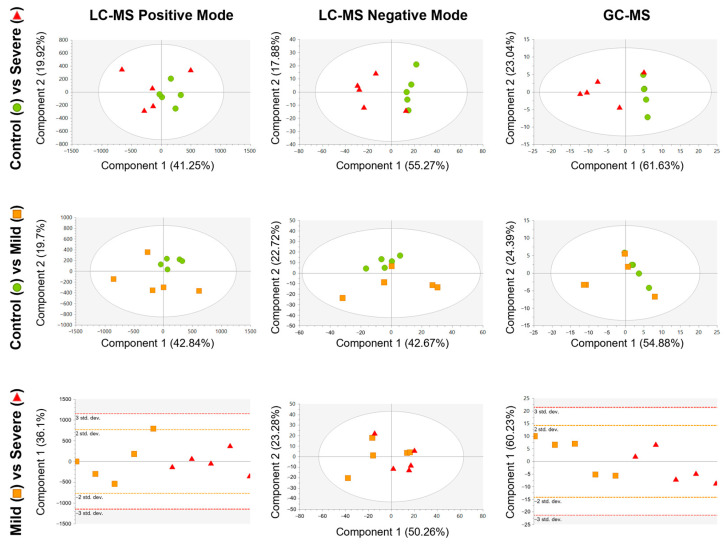
Score plots of the unsupervised PCA model for each comparison, built using the 242 signals for LC-MS positive mode (**left column**), 792 signals for LC-MS negative mode (**middle column**), and 38 signals for GC-MS (**right column**) that complied with quality criteria. Dots are coloured according to their group: control = green circle (*n* = 5); mild = orange square (*n* = 5); and severe = red triangle (*n* = 5). LC-MS in positive mode and GC-MS data were pareto scaled with no transformation, while LC-MS in negative mode data were unit variance scaled with no transformation.

**Figure 3 ijms-24-12714-f003:**
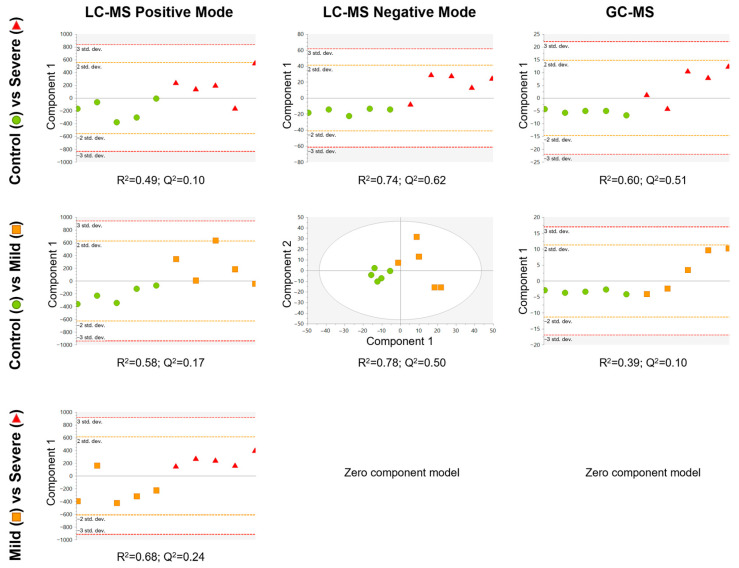
Supervised PLS-DA model for each comparison, built using the 242 signals for LC-MS positive mode (**left column**), 792 signals for LC-MS negative mode (**middle column**), and 38 metabolites for GC-MS (**right column**) that complied with quality criteria. Dots are coloured according to group: control = green circle (*n* = 5); mild = orange square (*n* = 5); and severe = red triangle (*n* = 5). LC-MS positive mode and GC-MS data were pareto scaled with no transformation, while LC-MS negative mode data were unit variance scaled with no transformation. R^2^ is the capability of the model to classify the samples; Q^2^ is the capability of the model to predict the class of a new sample.

**Figure 4 ijms-24-12714-f004:**
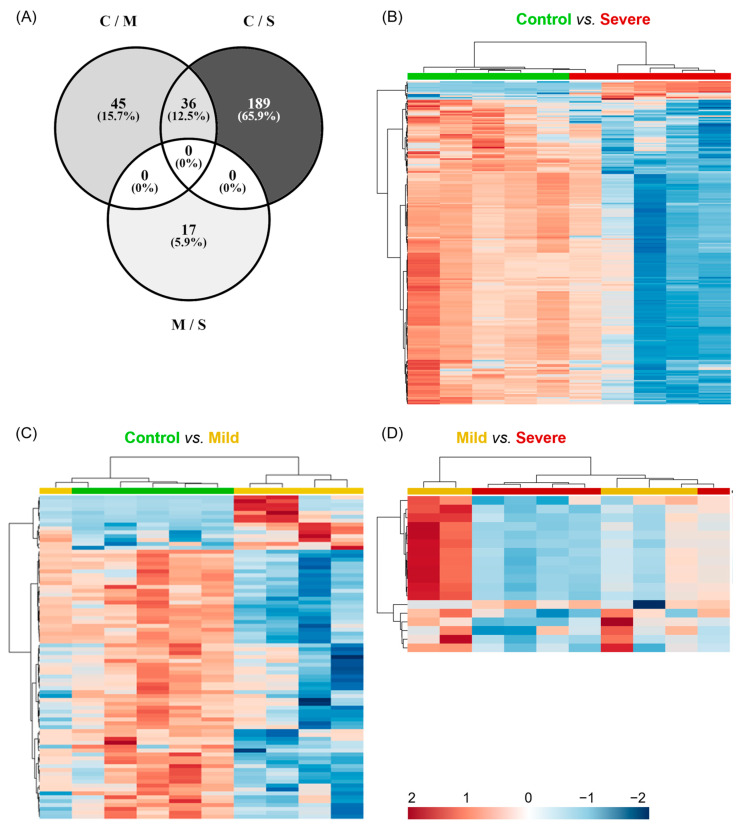
Metabolites differentially detected in PL-EVs cluster in severely allergic patients. (**A**–**C**) Representation of the significant chemical entities for each comparison detected in the metabolic profile of PL-EVs. (**A**) Venn diagram representing the distribution of 287 significant chemical entities (*p*-value < 0.05, *p*-FDR < 0.1) for each comparison (C; control; M, mild; S, severe). (**B**–**D**) Heatmaps with hierarchical clustering representing significant metabolomic signals. Samples (columns) and metabolites (rows) were grouped according to their similarity. Heatmaps were performed using MetaboAnalyst software (V 5.0.) with a Euclidean distance measure and Ward’s clustering method. Control is shown in green (*n* = 5), mild in yellow (*n* = 5), and severe in red (*n* = 5). Red cells indicate increased levels and blue cells indicate decreased levels. Detailed information about identified metabolites is available in Appendix A.

**Figure 5 ijms-24-12714-f005:**
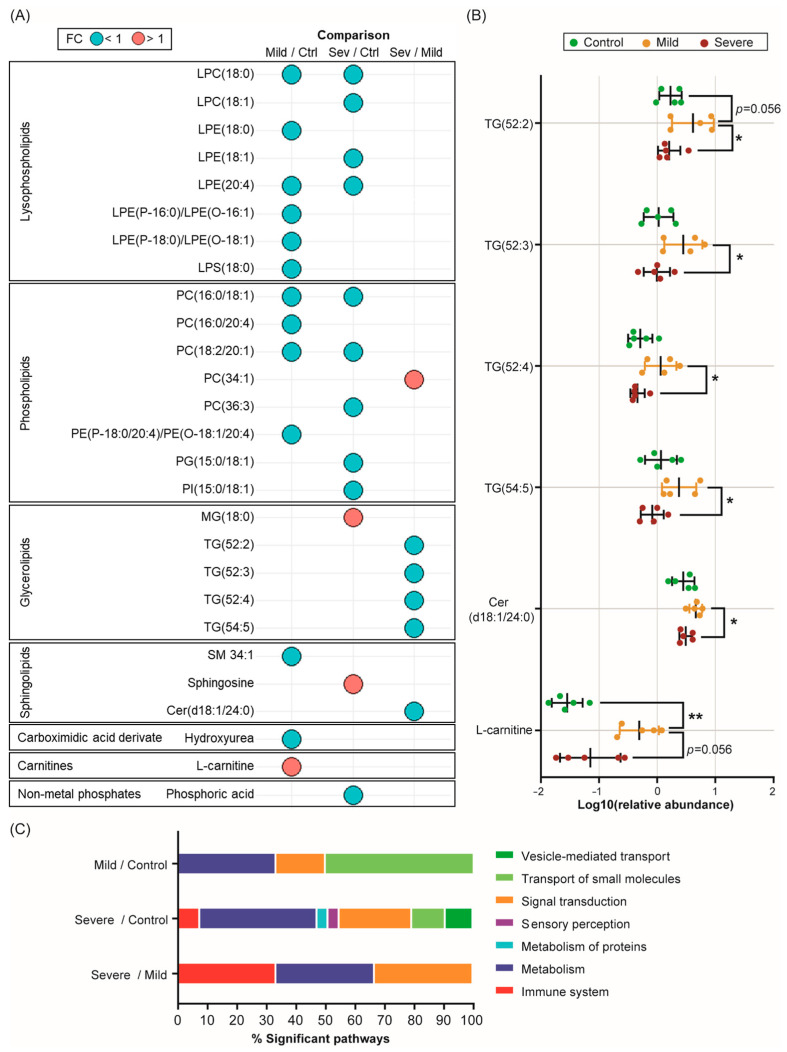
Severe allergic patients’ PL-EVs are loaded with specific metabolic mediators. (**A**) Bubble diagram representing the fold change (FC) of all significant (*p*-value < 0.05, *p*-FDR < 0.1) metabolites identified for each comparison. Red colour indicates a fold change greater than one and blue colour indicates a fold change of less than one. (**B**) Scatter plots showing the relative abundance (log10(relative abundance)) of the most relevant identified compounds (mean ± SD) in the PL-EVs obtained from control (green) subjects and mildly (yellow) and severely (red) allergic patients. The Mann–Witney U test was used to calculate significant differences. * *p* < 0.05; ** *p* < 0.01. Ctrl, control; Sev, severe. (**C**) Bar graph representation of the main categories of significant biological pathways (*p*-value < 0.05) involving statistically significant PL-EV metabolites from mildly allergic patients vs. controls, severely allergic patients vs. controls, and severely vs. mildly allergic patients’ comparisons.

## Data Availability

EV-TRACK ID: EV230007.

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
