# Peer review of "Platelet-Derived Extracellular Vesicles as Lipid Carriers in Severe Allergic Inflammation"

_ijms, 2023, doi:10.3390/ijms241612714_

Round 1
Reviewer 1 Report
The authors of the article ijms-2482534 entitled “Platelet-derived extracellular vesicles as lipid carriers in severe allergic inflammation", investigated differences in metabolites present in platelet derived extracellular vesicles, demonstrating a possible correlation of such changes in severe inflammation in severe allergic individuals. This is an interesting article which is very well written. However, there are a couple of concerns that need to be addressed.
1) Figure 5 and 6 could be merged into one with Figure 6 (panel A in new figure) actually preceding Figure 5 (panel B in new figure). There is no point these figures to be split.
2) The authors only include 6 patients from each group (and 8 controls) without having any kind of mechanistic/functional experiments of these metabolites (such as in vitro studies). It would be advised to smooth some sentences concerning the impact of their work throughout the text (such as using the word “might” in a few sentences).
No comment
Author Response
We want to thank the reviewer for their enthusiasm for our work and recognition of the novelty and importance of our findings. We have addressed each query as outlined in the point-by-point response uploaded as a PDF file.

Reviewer 2 Report
The original article draft title 'Platelet-derived extracellular vesicles as lipid carriers in severe allergic inflammation' presents a rigorous and well-conducted research work analyzing platelet-derived extracellular vesicles lipid content using liquid and gas chromatography coupled to mass spectrometry of mild and severe allergic patients and compare them to non-allergic controls. The authors performed a well-described work and analysis. Also, the results are well-described.
However, the next comments and questions need to be addressed to improve clarity:
1. Western blot presented in Figure 1G as 'representative' does not correspond to any of the three samples tested for the different purification methods (Figure S1). That is patent by the EV markers Alix and CD63. In Figure S1, fraction 1 from SEC+UF seems to have more Alix content. Although this does not compromise the rest of the work, please explain why none of the three samples included in the study was presented in Figure 1G.
2. In line 120, what does (NA-4 patient) means? If relevant, please include a clarification in the text.
3. In line 181, the number of significant differences in the comparison controls vs. severe allergic patients is 225, whereas the total number of significant differences in the study is 286, according to Figure 4A.
4. A whole paragraph describing the results of comparing controls vs. severe allergic patients in section 2.4 seems missing (before line 187).
5. In line 199, '… with a more pronounced reduction in the severe allergic group' is unclear. Please rewrite the sentence to improve clarity.
6. In line 215, the number of metabolites identified by LC-MS is 27, according to tables S6 and S7. Please correct the text and clarify in the text if these metabolites were identified with LC-MS.
7. In line 227, the FDR of L-Carnitine in the comparison of mild vs. severe allergic patients is 0.112, according to Table S7.
8. In Figure 5A, blue dots indicate that 'the metabolite is less abundant in the control or mild group,' whereas the expressed in the paragraph from lines 231-235 indicated that the amount of Cer(d18:1/20:4) and specific triglycerides (TGs) are less abundant in the severe allergic group. Please explain the discrepancies or correct the figure legend to 'the metabolite is less abundant than in the control or mild group.'
9. In the material and methods section, the methods referring to Size Exclusion Chromatography and ultrafiltration (SEC+UF) are unclear. Please rewrite this section.
10. Please correct line 35 from the supplementary materials.
11. In Figure S2, please correct the legend, as mild allergic patients are depicted in orange, not yellow.
Author Response

(The authors gave the same response as above.)

Reviewer 3 Report
The present article by Couto-Rodriguez determines the metabolic profile and changes in PL-EVs according to the degree of inflammation in allergic patients (mild and severe). While the study is interesting, there are some concerns that can be addressed to further improve the manuscript.
1. The characterization of the platelets EVs is unclear. The TEM images must incorporate arrows to point out the EVs in the image. 30k and 100k on the Y axis need to be defined.
2. The sample size is too small to draw a statistical conclusion in Figure 1, especially n=2 in Fig 1J, 1K is insufficient for statistical analysis. 3. In figure 2, why does the separation seem more contrasting in LC-MS negative mode only? The authors may explain LC-MS positive and LC-MS negative modes. 4. While it is not unsurprising to find that the identified metabolites were mainly lipids given their participation platelet activation, especially during platelet signaling. However, platelets also rely on glycolysis for their energy requirements. Did the authors identify any glycolytic intermediates? Minor comments: 1. The western blots are not shown appropriately in figure 1. The CD61 blot crops into the bands and require replacement. 2. Did the authors look at the metabolic profile of platelets in control vs inflammation (severe and mild) before investigating the same in platelets EVs? It seems logical to explore this first in platelets before investigating the same in EVs.
English is fine and easily understandable.
Author Response

(The authors gave the same response as above.)

Reviewer 4 Report
The authors investigated the lipid and metabolite contents in extracellular vesicles (EVs) isolated from the plasma-rich plasma (PRP) control donors, mild and severe allergic patients. They report no changes in the levels and size of EVs (mainly platelet-derived EVs, PL-EVs) between the three groups. However, their report that the EV lipid and metabolite contents varied according to allergy severity with generally more pronounced decreases in patients with severe allergic inflammation.
General contents: The manuscript brings confusion, and several issues require clarification. First, the two methods used to purify EVs from PRPs do not distinguish the EV subtypes (platelets, red blood cells, white blood cells). The authors assume they are PL-EVs, but it is partly true. At best, the study is a characterization of plasma EVs from control donors and allergic patients since there is no quantification of PL-EVs vs. EVs originating from other blood cells. The authors should change the title of their manuscript and refer to EVs, not PL-EVs.
Platelet-derived EVs are heterogenous and produced through distinct pathways (secretion of exosomes, budding from the plasma membrane). The few markers used in this study (CD81, CD9, CD63, ALIX) are essentially exosome markers. There is no attempt to use a protein marker (CD41) for assessing the levels of plasma membrane-derived EVs in the samples.
The discussion and the conclusions are too speculative. There is no serious discussion of the limits and weaknesses (small number of patients, high inter-sample variability, EV subtypes).
Minor point:
Line 66, PMA? I guess the authors mean platelet-monocyte aggregates (PMA).
Moderate editing of the English is required.
Author Response

(The authors gave the same response as above.)

Round 2
Reviewer 4 Report
The authors performed proteomic, metabolomic, and lipidomic analyses of extracellular vesicles (EVs) isolated from apheresis platelet concentrates [also named plasma-rich plasma (PRP)] of controls, mild and severe allergic patients. The authors report no change in the levels and size of platelets EVs (PL-EVs) between various groups. In contrast, EV lipids and metabolites differed amongst the three groups, with more pronounced alterations in patients with severe allergic inflammation.
General comments: I thank the authors for their reply to previous criticisms. I'm partially satisfied with these responses. In addition, the author's reply to the point raises other issues.
First, the authors must add sentence(s) saying that accurate measurements of plasma EVs depend on various variables, including blood collection, anticoagulant, plasma preparation, and sample isolation. Flow cytometry is most used for qualitative and quantitative measurement of EVs in PPPs as it prevents the potential loss associated with EV isolation. Different isolation analysis methods could explain the discrepancies with other studies (Ref. 29-32). The traditional way to prepare EVs is sequential centrifugation of blood and PRP at low speed to obtain PPP. To my understanding, the platelet count in PRP obtained by sequential centrifugation of blood samples is 4 to 6-fold lower than that of PRP samples collected by apheresis.
Second, I am puzzled by the following statement made by the authors:
“We isolated EVs by ultracentrifugation from both PRP and PPP and used protein quantification as a measure of EV presence. Our results confirmed that we obtained between 300 and 1000 µg/ml of protein when isolating EVs from two mL of PRP. In contrast, there was no protein obtained after the isolation of EVs from PPP, meaning that PPP does not contain detectable EVs.”
The point is that several studies have identified and quantitated EVs in PRP using high-sensitivity flow cytometry. Furthermore, the results suggest that EVs are produced in the time interval between apheresis and EV isolation from platelet concentrate samples. I won't comment further on information not included in the article. I want to draw the author's attention to this point as accurate measurements of plasma EVs depend, in part, on plasma preparation and sample isolation (see first comment above).
I respectfully highlight that various cells express CD61, including leukocytes and endothelial cells. I understand that most circulating CD61-positive EVs originate from platelets, but the author cannot dismiss other sources of CD61-positive EVs. The addition of the sentence is necessary to strengthen the discussion.
Line 221: Please replace phosphocholine with phosphatidylcholine.
Author Response
We would like to thank the Reviewer for taking the time and effort to review the manuscript. We are very grateful for Reviewer’s valuable and constructive comments and suggestions. We have checked carefully and revised our manuscript based on the comments provided. The specific questions were answered in the attached document. We hope that the revised manuscript is now suitable for your further consideration.
